# Semantic Image Manipulation with Background-guided Internal Learning

## Abstract

Image manipulation has attracted a lot of interest due to its wide range of applications. Prior work modifies images either from pixel-level manipulation, such as image inpainting or through manual edits via paintbrushes and scribbles, or from high-level manipulation, employing deep generative networks to output an image conditioned on high-level semantic input. In this study, we propose Semantic Image Manipulation with Background-guided Internal Learning (SIMBIL), which combines high-level and pixel-level manipulation. Specifically, users can edit an image at the semantic level by applying changes on the scene graph. Then our model manipulates the image at the pixel level according to the modified scene graph. There are two major advantages of our approach. First, high-level manipulation requires less manual effort from the user compared to manipulating raw image pixels. Second, our pixel-level internal learning approach is scalable to images of various sizes without reliance on external visual datasets for training. We outperform the state-of-the-art in a quantitative and qualitative evaluation on CLEVR and Visual Genome datasets. Experiments show around 8 points improvement of SSIM (RoI) on CLEVR and we found human users preferred our manipulated images over prior work by 9-33% on Visual Genome, demonstrating the effectiveness of our approach.

## 1 Introduction

Image manipulation modifies the content of an image according to user guidance. The task can be solved in two primary ways: pixel-level manipulation on raw images and high-level manipulation on image semantics. Pixel-level manipulation spans image inpainting (Zhao et al., 2019; Yeh et al., 2017), colorization (Zhang et al., 2016), object removal (Shetty et al., 2018), style transfer (Gatys et al., 2016), image extension (Teterwak et al., 2019), etc. Pixel-level manipulation methods do not need to understand the semantic meanings of an image. In contrast, high-level manipulation often uses deep generative networks conditioned on user inputs like semantic maps and language descriptions to identify the desired modifications. Most prior work for high-level image manipulation are object-centric, such as human face transfer (Choi et al., 2018; Lee et al., 2020; Jo & Park, 2019; Zhao et al., 2018) and object appearance or attribute modification (Li et al., 2020a; Liang et al., 2018). Recently, approaches modifying the entire scenes by instance maps (Wang et al., 2018), language descriptions (El-Nouby et al., 2019; Nichol et al., 2021; Avrahami et al., 2022) or scene graphs (Dhamo et al., 2020) are also proposed. Although high-level manipulation requires less manual effort from users, deep generative networks for high-level manipulation have two drawbacks. First, high-level manipulation frameworks often only support outputting low-resolution images due to GPU memory requirements (Dhamo et al., 2020). Super-resolution modules (Saharia et al., 2022; Nichol et al., 2021) are required to get higher-resolution images, introducing extra overhead. Second, generative models may result in the loss of attributes and details of the original images (Bau et al., 2020).

Ideally, a good image manipulation method should satisfy the following requirements: (1) provide maximum convenience to users; for example, manipulating images by scene graphs or language description is more convenient than manually segmenting, replacing, or removing the target object, (2) preserve the textures and details of the original image in appropriate areas, (3) correctly modify the target region of the image according to user instructions, (4) ability to generalize across input images without relying on specific external datasets. There are two major challenges to developing an approach that can satisfy these requirements. First, it is challenging for existing text-driven

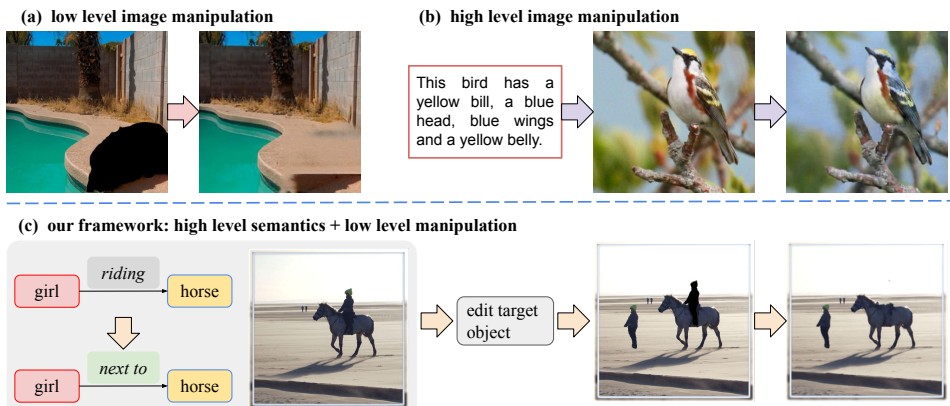

Figure 1: Prior work of image manipulation is either at pixel-level (*e.g.*, EdgeConnect (Nazeri et al., 2019)), shown in (a), or high-level (*e.g.*, ManiGAN (Li et al., 2020a)), shown in (b). In our work, shown in (c), we address the issues of prior work (see Section 1 for discussion) by connecting high-level semantics with pixel-level manipulation, where the semantic level information is encoded by an RNN-based scene-graph encoder. Then the pixel-level manipulation, background-guided internal learning, is done according to the processed information.

image editing methods to accurately localize the Region of Interest (RoI)[1] at complex scenes. *E.g.*, popular frameworks including GLIDE (Nichol et al., 2021) and blended diffusion model (Avrahami et al., 2022) require users to manually select RoI. Methods (Li et al., 2020a;b) that do not require bounding boxes as input are mostly object-centric and the images do not contain complex semantic relationships between objects. The ambiguity of text makes developing an RoI prediction model challenging. For example, if there are multiple birds in an image, locating the target bird according to text descriptions would be challenging even for a human. To solve this issue, we use scene graph information to eliminate the ambiguity of text, while still making manipulations easy. Second, most image inpainting methods (Nazeri et al., 2019; Yu et al., 2019; Rombach et al., 2022) trained their models based on reconstruction task. In this case, as we will show in Section 4.2 and Section 4.3, these external learning methods tend to repair the target object even if the user command is to remove the object. We further introduce internal learning to avoid the object repair issue.

Specifically, we propose a Semantic Image Manipulation framework with Background-guided Internal Learning (SIMBIL). SIMBIL combines high-level image semantics with pixel-level manipulation. Figure 1 illustrates the difference between SIMBIL and prior work by an object relationship change example. Figure 2 presents the overall structure of SIMBIL. First, the target object is determined by the scene graph of an image. The users are able to edit the nodes and edges of scene graphs for four operations, object removal, object replacement, semantic relationship change, and object addition. We use a segmentation module to outline the mask of target object. A Recurrent Neural Networks (RNN)-based module further encodes the semantic modifications between the objects and predicts the target Region of Interest (RoI) according to editing operations. Finally, we improve Deep Image Prior (DIP) (Ulyanov et al., 2018) by utilizing background pixels as a constraint and propose the background-guided internal learning module.

In summary, the contributions of this paper are:
- We propose a semantic image manipulation framework (SIMBIL) to combine high-level semantics with pixel-level image manipulation, reducing manual effort and alleviating the issues caused by prior work. Notably, compared to existing manipulation methods using scene graphs (Dhamo et al., 2020), SIMBIL can generate higher resolution images while accurately preserving the original details of the input images.
- We develop a background-guided internal learning algorithm based on DIP (Ulyanov et al., 2018) for image inpainting, which utilizes the average value of the background pixels around the missing part as guidance as opposed to only relying on the implicit prior captured by the neural network parameterization, boosting performance.

---
[1]We use RoI to indicate the region that is supposed to be edited in the image.

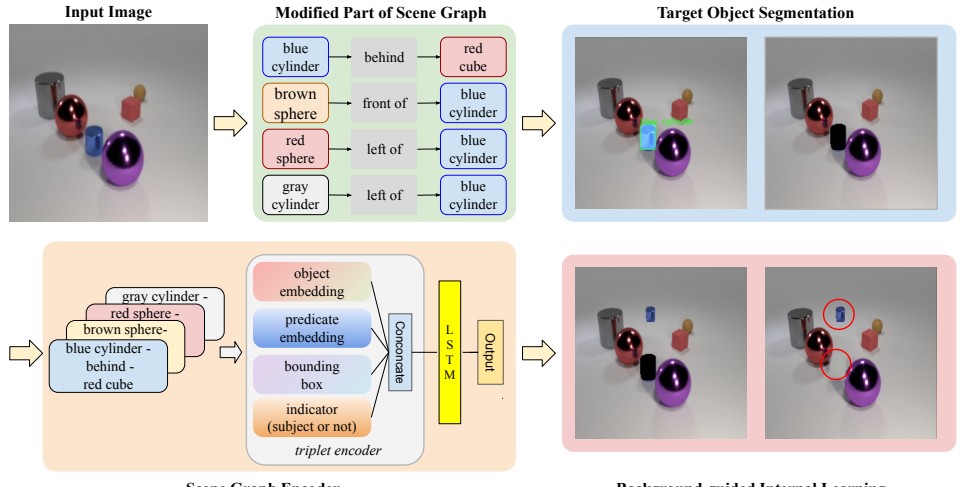

Figure 2: **SIMBIL overview.** Given an input image and its modified scene graph, our approach consists of three modules in sequence: (1) Object Segmentation Module: segmenting the target object according to the modified scene graph; (2) RoI Prediction Module: predicting the Region of Interst by encoding the modified triplets of the scene graph; (3) Internal Learning Module: leveraging the average value of the background pixels around the target object to guide the inpainting results.

- Quantitative and qualitative experiments on CLEVR (Johnson et al., 2017) and Visual Genome (Krishna et al., 2017) demonstrate SIMBIL outperforms the state-of-the-art.
- Extensive experiments on high resolution images demonstrate SIMBIL's flexibility and scalability.

## 2 RELATED WORK

**Image Manipulation**. Many studies based on image synthesis focus on object-centric scenarios, *e.g.*, editing human face (Choi et al., 2018; Lee et al., 2020; Jo & Park, 2019; Yeh et al., 2017; Zhao et al., 2018), text-guided attribute manipulation (Liang et al., 2018; Li et al., 2020a), manually editing with paintbrush and scribbles (Brock et al., 2016; Zhu et al., 2016). Recently there has been some work modifying images in complex scenes consisting of multiple objects (Wang et al., 2018; El-Nouby et al., 2019; Tan et al., 2019; Dhamo et al., 2020; Nichol et al., 2021; Avrahami et al., 2022). These methods often require human effort to outline the region of interest (Wang et al., 2018; Nichol et al., 2021; Avrahami et al., 2022) and focus only on object replacement or addition (El-Nouby et al., 2019; Tan et al., 2019; Nichol et al., 2021; Avrahami et al., 2022). Dhamo et al. (2020) proposed a scene-graph based approach that can also edit the semantic relationships between objects in a single image. Unlike methods based on generative networks, SIMBIL adopts an automatic "photoshop" mechanism, directly manipulating raw pixels. Our method is applicable to different input images while preserving the original image details.

**Internal Learning.** While external learning trains a model on external datasets, internal learning methods such as Deep Image Prior (Ulyanov et al., 2018) use a generator network trained on a single image to address tasks like image denoising, inpainting, and super-resolution. Internal learning is not limited to image inverse problems (Guasch et al., 2020; Zhang & Lin, 2020; Zhang et al., 2019b), but has other applications including video motion transfer (Chan et al., 2019), video inpainting (Zhang et al., 2019a), and semantic photo manipulation (Bau et al., 2020). Gandelsman et al. (2019) also applied coupled DIPs to unsupervised image decomposition. In this paper, we introduce a background-guided mechanism based on DIP, which uses the background pixels for more accurate guidance than the implicit prior used by DIP for the missing region of an image.

**Scene Graphs and Visual Relationship Detection.** Scene graphs (Johnson et al., 2015) describe the objects, attributes of objects, and relationships between objects in an image, and methods that generate them can be divided into two categories: Convolutional Neural Network (CNN)-based methods (Li et al., 2017; 2018; Yang et al., 2018; Qi et al., 2019) and Recurrent Neural Networks

(RNN)-based methods (Herzig et al., 2018; Xu et al., 2017; Zellers et al., 2018). At a high level, scene graph construction combines object/entity detection (Ren et al., 2015; Plummer et al., 2020) and detecting their visual relationships (Dai et al., 2017; Lu et al., 2016; Plummer et al., 2017).

In our paper, users can apply changes to the relationships between objects to realize image manipulation. A relevant task is visual entity localization according to visual relationships. Krishna et al. (2018) introduce an iterative model to localize the entities in the referring relationship. Plummer et al. (2017) combine linguistic cues with learned weights for phrase localization. In our project, the main difference is that the target object is invisible in the image. Therefore, we propose an RNN-based method to automatically predict a plausible position for the target object according to the existing information from the image and `<subject-predicate-object>` triplets.

## 3 SIMBIL

Given an input image $I$ and its corresponding scene graph $G$, users can apply changes on $G$. Our task is to perform semantic manipulation according to the user-modified scene graph $\widetilde{G}$ and the original input image $I$. In practice, $G$ can be obtained by scene graph generation methods (Li et al., 2018; Qi et al., 2019; Herzig et al., 2018; Xu et al., 2017; Zellers et al., 2018). Following (Dhamo et al., 2020), we use ground truth scene graphs in our experiments so the accuracy of current methods is not a factor in evaluating our approach. Figure 2 provides an overview of our framework. We introduce four kinds of modifications on scene graphs in Section 3.1. In Section 3.2, we discuss the details of our RoI prediction model. The background-guided internal learning algorithm is then presented in Section 3.3. Finally, we summarize the overall learning strategy in Section 3.4.

### 3.1 MODIFICATIONS ON SCENE GRAPHS

Following (Dhamo et al., 2020), we perform four tasks of semantic manipulation: object addition, object replacement, relationship change, and object removal[2]. These manipulations are reflected on the nodes and edges of scene graphs. Take the blue cylinder in Figure 2 as an example, (1) Object Addition: adding a new object (node) and its corresponding relationships (edges) with other objects on the scene graph; (2) Object Replacement: replacing the node which represents `<blue cylinder>` to another object; (3) Relationship Change: changing one spatial relationship of blue cylinder (edge) from `<blue cylinder-front of-red cude>` to `<blue cylinder-behind-red cude>`. Other relationships of the blue cylinder can be changed similarly; (4) Object Removal: deleting the node of `<blue cylinder>` and its corresponding edges. Given the modified scene graph $\widetilde{G}$, we extract the bounding box and mask of the target object by MaskFormer (Cheng et al., 2021), and then proceed with performing the alteration.

### 3.2 REGION OF INTEREST (RoI) PREDICTION

We introduce a RoI prediction module to support the object addition and semantic relationship change tasks. Benefiting from this module, SIMBIL can automatically predict RoI instead of manually outlining the bounding box (Nichol et al., 2021; Avrahami et al., 2022). Unlike relationship detection (Krishna et al., 2017), which identifies relationships between a visible reference and target object in a scene, our goal is to determine where to place the target object given the reference object and its semantic relationship to the target object. In addition, multiple relationships may need to be encoded as the target object may have relationships with multiple objects in the scene. Inspired by RNN-based methods on scene graph generation (Herzig et al., 2018; Zellers et al., 2018), we develop an LSTM-based model that encodes all the triplets of the modified scene graph.

We define the categories of the entities (subject & object) in images as $O = \{o_1, ..., o_n\}$, the corresponding bounding boxes as $B = \{b_1, ..., b_n\}$, and the relationships (predicates) between entities as $R = \{r_1, ..., r_m\}$. Given the modified scene graph $\widetilde{G}$ and the target object $o$ in an image, the triplets of the target object are referred as $\mathbf{y} = \{y_1, ..., y_T\}$. For $t \in \{1, ..., T\}$, $y_t$ is a triplet in $< s - p - o >$ format, where $s, o \in O$ and $p \in R$. For each triplet $y_t$, we devise two embedding layers that obtain the object embedding $V_s$, $V_o$ and predicate embedding $V_p$ separately. We

---

[2]Following Dhamo et al. (2020), we perform four operations on CLEVR (Johnson et al., 2017), three operations on Visual Genome (Krishna et al., 2017): object replacement, relationship change, and object removal.

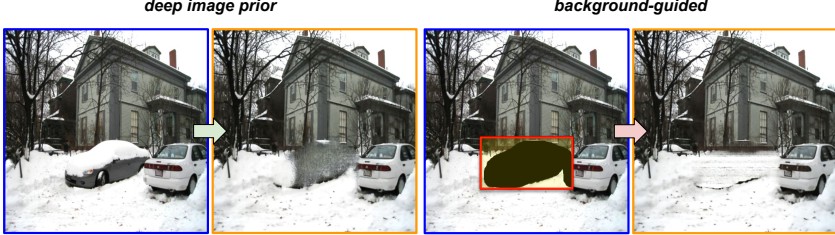

*deep image prior*           *background-guided*

Figure 3: Left: deep image prior; Right: background-guided internal learning. Our methods generate more reasonsble inpainting result by leveraging the background pixels as guidance.

also introduce a binary indicator $I$ to indicate whether the target object is subject or object in $y_t$. Suppose the reference object corresponds to "subject" and the target object corresponds to "object", we also consider the position of reference object $b_s$ as part of the input to our model. The concatenation of these features are encoded by a LSTM model. Specifically, it can be expressed as: $x_t = concat\{V_s, V_o, V_p, b_s, I\}; h_t = LSTM(x_t, h_{t-1})$, where $h_t$ is the hidden state of LSTM at triplet $y_t$. $h_T$ is followed by an MLP to predict the RoI. We crop the final output to range (0∼1) since the image size is normalized and train the model using Mean squared error (MSE).

### 3.3 BACKGROUND-GUIDED INTERNAL LEARNING

Our internal learning approach, SIMBIL, has two advantages over external learning methods (Nazeri et al., 2019; Yu et al., 2019; Rombach et al., 2022). First, external learning methods rely on the training data that consists of image pairs before and after inpainting. Thus, an inpainting model for CLEVR and Visual Genome needs to be trained separately and is not applicable to different input images. Second, as we will show in Section 4, many external learning methods tend to repair rather than remove objects. This is because these models use the ground truth image instead of the modified image as supervision (*i.e.*, Visual Genome lacks before and after editing image pairs).

Given an image $x_0$ and a binary mask $m$, the input to image inpainting is $x_0 \odot m$, where $m$ defines the known areas of image $x_0$ (1 for the known regions and 0 otherwise) and $\odot$ denotes Hadamard's product. The standard image inversion problem can be formulated by

$$x^* = \underset{x}{\operatorname{argmin}} \, E(x; x_0) + R(x) \tag{1}$$

where $E(x; x_0)$ is a task-dependent data term and $R(x)$ is the regularization term. For deep image prior (DIP) (Ulyanov et al., 2018), the regularization term $R(x)$ is replaced by the implicit prior from the neural network parametrization. Specifically, the input of the generative neural network is a noise map $z$ that has one channel and shares the same spatial size as image $x_0$. The network parameters are updated to minimize $E(x; x_0)$. In terms of image inpainting, $E(x; x_0)$ satisfies

$$E(x; x_0) = ||(x - x_0) \odot m||^2 \tag{2}$$

where $x$ is the output of the generative model. Note that our "missing" regions are determined by a segmentation model, *i.e.*, the boundary of the hole might not be accurate. Thus, the generative network may be given a noisy input, resulting in incorrect predictions by prior work like DIP. In addition, the object to remove can be large, in which case the relative performance of DIP is worse (Ulyanov et al., 2018; Zhang et al., 2019a).

We address the aforementioned issue with noisy inputs in two ways. First, we use mask dilation to alleviate the misinformation of the boundary. Second, we use the background pixels around the hole as guidance to the model. Specifically, suppose a consistent background is divided into two parts; the distributions of each part should still be consistent with the other. As shown in Figure 3, the segmentation of the target car is the missing region of the original image. Then the background pixels within the red bounding box are extracted from the image. We constrain the average of the missing region by the average of the specified background region. Denote the average value of background pixels as $B, B \in \mathbb{R}^3$. our new objective function is:

$$E(x; x_0) = ||(x - x_0) \odot m||^2 + \lambda \frac{1}{C} ||average(x \odot (1 - m)) - B||^2, \tag{3}$$

where $C$ represents the number of image channels, and $\lambda$ is the hyper-parameter to control the two loss terms. In practice, the second loss term can be further divided into sub-terms, for example, constraining the average row by row.

### 3.4 OVERALL LEARNING STRATEGY

There are three components in our framework, which are segmentation module, RoI prediction module, and background-guided internal learning. The segmentation module (Section 3.1) is applied in two cases (1) removing the target object from the original position, (2) searching relevant objects (from the current image or query images) given the object category, which is mainly for object addition and replacement[3]. The RoI prediction model (Section 3.2) is designed for object addition and relationship change. To train this model, we extract modified triplets, bounding boxes of reference objects, and target objects to construct a dataset. Take CLEVR as an example; we extract 6688 images from the training set to construct the training data, then the 818 images in the validation set are used to evaluate the performance of the model. The target object is pasted according to the predicted RoI of the model. After this step, remember we still need to address the missing region ("hole"), which is accomplished using our proposed background-guided internal learning (Section 3.3) to get the final output. More implementation details are provided in Appendix A.

## 4 EXPERIMENTS

### 4.1 DATASETS AND EXPERIMENT SETTINGS

**Datasets.** We evaluate on CLEVR (Johnson et al., 2017) and Visual Genome (Krishna et al., 2017). CLEVR is a synthetic dataset that contains ground truth pairs for image editing. For a fair comparison, we use the test set provided by (Dhamo et al., 2020). Visual Genome, on the other hand, lacks before and after image editing pairs. Therefore, we leverage human evaluation to estimate the correctness of manipulation. Qualitative examples from both CLEVR and Visual Genome are presented to demonstrate the effectiveness of our model on synthetic and natural images.

**Metrics.** Following Dhamo et al. (2020), we report mean absolute error (MAE) and structural similarity index measure (SSIM) of RoI to evaluate the model. We defined the modified area of images as RoI so that it can directly reflect the manipulation accuracy. In addition, we also performed user study to evaluate whether the edited images are consistent to editing commands.

**Baselines.** We use the code and pretrained model of SIMSG (Dhamo et al., 2020) as our scene-graph-guided image editing baseline. In addition, we apply baselines on separate tasks. Specifically, we use DeepFillv2 (Yu et al., 2019), EdgeConenct (Nazeri et al., 2019), and Latent Diffusion Model (LDM) (Rombach et al., 2022) on object removal, use Blended Diffusion Model (BDM) (Avrahami et al., 2022) on object replacement and addition, use GLIDE (Nichol et al., 2021) on object removal, replacement, and addition. Since GLIDE and BDM require manually outline RoI, we apply RoI prediction of SIMBIL to these methods.

### 4.2 PERFORMANCE ON CLEVR DATASET

**Manipulation Experiments.** Table 1 reports quantitative results of different methods at $256 \times 256$[4] image resolution. From the table, we see that SIMBIL notably outperforms baselines on four image editing tasks, especially for object removal and relationship change. We draw three major conclusions from Table 1. First, external image inpainting approches (Yu et al., 2019; Nazeri et al., 2019; Rombach et al., 2022) do not perform well on our object removal task. As we discussed in Section 3.3, these methods tend to introduce unexpected objects to repair the missing regions. Second, scene-graph based image editing approaches, SIMSG and SIMBIL, perform more accurate manipulation operations on object replacement and object addition compared to text-driven methods GLIDE and BDM. This is because scene graph contains semantic relationships and object positions in the

---

[3]We adopt compositional scene representation mechanism for object addition and replacement following Text2Scene (Tan et al., 2019)

[4]Since SIMSG requires more than 48GB GPU memory to train a model that outputs $256 \times 256$ images, we apply USRNet (Zhang et al., 2020) on the output of SIMSG to avoid out-of-memory issue.

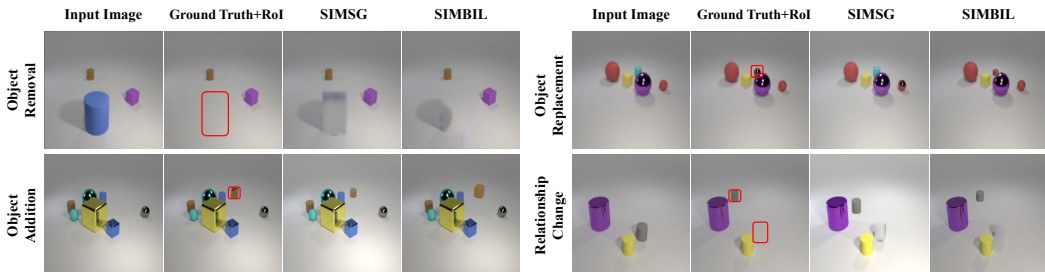

Figure 4: Comparing SIMBIL to SIMSG on CLEVR. RoI is outlined by red bounding boxes. See Section 4.2 for discussion.

Table 1: Quantitative results of image manipulation on CLEVR dataset. Empty value means the corresponding approach is not capable of this task. See Section 4.2 for discussion.

| Method | Object Removal | | Object Replacement | | Object Addition | | Relationship Change | |
|---|---|---|---|---|---|---|---|---|
| | MAE ↓ | SSIM ↑ | MAE ↓ | SSIM ↑ | MAE ↓ | SSIM ↑ | MAE ↓ | SSIM ↑ |
| DeepFillv2 (Yu et al., 2019) | 39.77 | 65.10 | - | - | - | - | - | - |
| EdgeConnect (Nazeri et al., 2019) | 29.26 | 66.70 | - | - | - | - | - | - |
| LDM (Rombach et al., 2022) | 37.84 | 52.41 | - | - | - | - | - | - |
| LAMA (Suvorov et al., 2022) | 33.54 | 71.19 | - | - | - | - | - | - |
| MAT (Li et al., 2022) | 31.46 | 74.80 | - | - | - | - | - | - |
| BDM (Avrahami et al., 2022) | - | - | 42.32 | 50.12 | 59.25 | 52.64 | - | - |
| GLIDE (Nichol et al., 2021) | 39.51 | 62.91 | 34.59 | 55.57 | 40.25 | 59.04 | - | - |
| SIMSG (Dhamo et al., 2020) | 30.38 | 85.89 | 33.76 | 67.41 | 44.58 | **65.80** | 33.31 | 85.95 |
| SIMBIL (ours) | **9.90** | **93.33** | **27.64** | **68.05** | **34.18** | 64.78 | **11.40** | **92.32** |

image which are difficult to be described by text guidance. The relevant information is important for models to accurately predict manipulated images. Third, we observe that though SIMSG achieves comparable SSIM scores with our method on object replacement and addition, the MAE loss of SIMSG is much higher than our model, which means the original attributes such as brightness are likely to be modified by SIMSG. To validate our analysis, we present images of different manipulations in Figure 4. The figure shows that SIMBIL can accurately edit images while still preserving the original background perfectly. In contrast, the brightness of SIMSG output is different from original images (*e.g.*, output of object addition & relationship change), resulting in higher MAE loss.

### 4.3 PERFORMANCE ON VISUAL GENOME DATASET

**Manipulation Experiments.** As with experiments on CLEVR, we apply popular inpainting methods on object removal, text-driven editing methods on object replacement, and SIMSG on all manipulation tasks. For each image editing task, we randomly selected 30 images generated by each baseline. This results in 150 images for object removal, 120 images for object replacement, and 60 images for relationship changes. Each image is annotated three times by AMT workers and we asked our annotators to judge whether the image is correctly manipulated according to the input guidance. In Table 2, we report that SIMBIL significantly outperforms baselines, especially for object removal and relationship change tasks, which is consistent with our conclusions on CLEVR.

We provide qualitative results in Figure 5. We set the image resolution to $256 \times 256$ to fit the output for most baselines. For SIMSG, we apply USRNet to increase the resolution from $64 \times 64$ to $256 \times 256$. Figure 5 (a) provides object replacement results, where SIMBIL clearly outperforms both scene-graph driven approach SIMSG and text-driven approaches GLIDE and BDM. Though BDM and GLIDE can generate some plausible objects in some cases, the appearance and shape of the edited object still look odd. Figure 5 (b) presents images reports by relationship change. From these results, we can see that our RoI prediction module outputs reasonable values for the relationship change between objects, for instance, riding → beside, near → on. Figure 5 (c) provides object removal examples. Consistent with our discussion in Section 3.3 and Section 4.2, external approaches tend to introduce unexpected objects in some cases when inpainting. We believe the

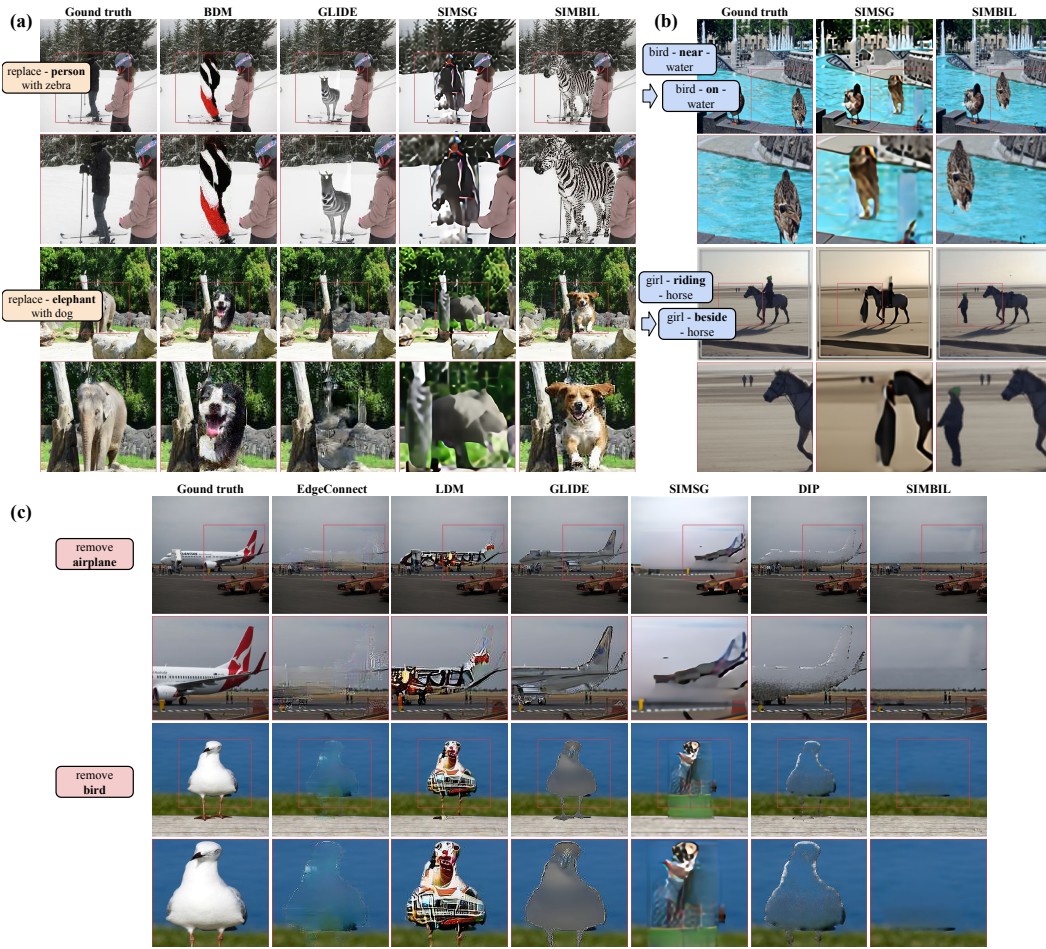

Figure 5: Qualitative results comparing SIMBIL with baselines on Visual Genome. (a): object replacement; (b): relationship change; (c): object removal. The edited regions are zoomed in below each image. See Section 4.3 for discussion.

Table 2: User evaluation judging the correctness of an image manipulation on Visual Genome. See Section 4.3 for discussion.

|  | Object Removal | Object Replacement | Relationship Change |
|---|---|---|---|
| EdgeConnect (Nazeri et al., 2019) | 27.8% | - | - |
| Latent Diffusion (Rombach et al., 2022) | 21.1% | - | - |
| Blended Diffusion (Avrahami et al., 2022) | - | 31.1% | - |
| GLIDE (Nichol et al., 2021) | 23.3% | 28.8% | - |
| SIMSG (Dhamo et al., 2020) | 22.2% | 24.4% | 15.6% |
| SIMBIL(ours) | **50.0%** | **40.0%** | **48.9%** |

model may recognize the appearance of some objects due to seeing similar images during training. In contrast, internal learning approaches do not suffer from this issue because they only utilize the context and textures of the current input image. Additionally, our background-guided method provides higher quality images compared to DIP.

## 4.4 LIMITATIONS AND FUTURE WORK

In this section, we analyze the limitations of our method and provide suggestions to address these limitations. Four cases are presented in Figure 6.

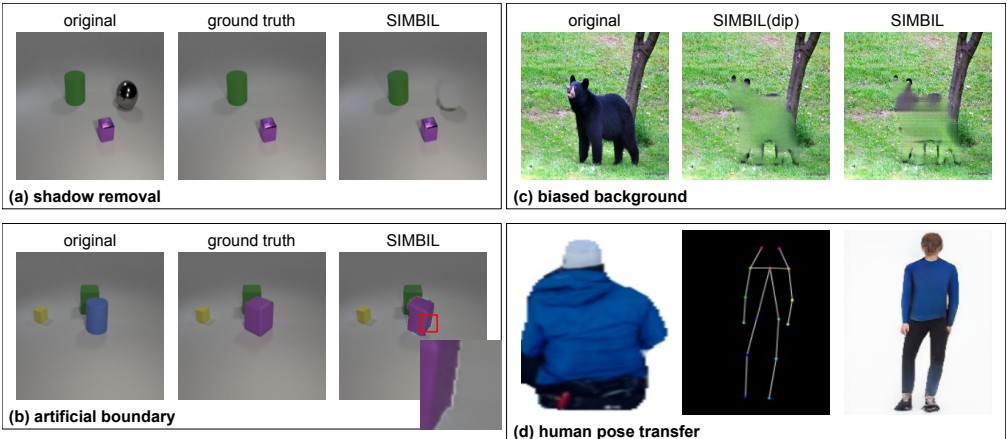

Figure 6: Limitations of SIMBIL. See Section 4.4 for discussion.

(a) **Shadow removal**: Since our segmentation module does not consider shadows as part of an object, they typically remain in an image. However, this can be addressed by incorporating shadow detection methods (*e.g.*, (Wang et al., 2020)) in the removal module.

(b) **Artificial boundary**: In some images, we can observe white edges if we zoom out the boundary region of the modified object. We assume the artificial boundary is introduced when we resize the object from the query images.

(c) **Biased background**: Although background-guided internal learning outperforms DIP in most cases, we do have some failures like the example in Figure 6(c). We believe this is because our segmentation model does not distinguish the tree from the background. As a result, our approach guides the missing region to be similar to the tree in some areas within the "hole".

(d) **Human pose transfer**: Relationships between objects are not limited to spatial relationships such as beside, near, on, but also span predicates like standing on, running, walking. Ideally, the appearance of the objects should also be changed according to various predicates. We tried to apply a human pose transfer model (Zhu et al., 2019) pretrained on DeepFashion (Liu et al., 2016) to address this issue. However, there is a significant domain shift from DeepFashion to Visual Genome that resulted in poor performance, which could be addressed in future work via domain adaptation techniques.

(e) **Inference time:** Internal learning methods train a separate network to process each image. The inference time to process a $256 \times 256$ image by a RTX 3090 takes approximately 35 seconds. We observe that our background-guidance mechanism has negligible overhead compared to DIP. For comparison, text-driven methods BDM (Avrahami et al., 2022) takes approximately 28 seconds and GLIDE (Nichol et al., 2021) takes approximately 8 seconds to process each image. While our approach does take a little longer at inference time, it does not have the expensive training step (in terms of data and computational time) of BDM or GLIDE.

## 5 CONCLUSION

In this paper, we proposed a semantic image manipulation method called SIMBIL that combines high-level image semantics with pixel-level manipulation. SIMBIL mainly consists of object segmentation module, an RNN-based RoI prediction module to predict the edited region for target objects, and a background-guided internal learning module for image inpainting. SIMBIL outperforms the state-of-the-art in a qualitative and quantitative evaluation on CLEVR and Visual Genome datasets. For example, our method outperforms SIMSG around 8 points improvement of SSIM on CLEVR and around 25% improvement of user evaluation accuracy on Visual Genome. Extensive experiments on images with higher resolution, which prior work struggled to perform, further demonstrate the effectiveness of our method. Thus, we argue the combination of high-level semantics and pixel-level manipulation is a promising way to solve the image manipulation problem since it requires less human effort and accurately preserves the original image's attributes and details.

## 6 REPRODUCIBILITY STATEMENT

Our method, SIMBIL, is presented in detail in Section 3. Figure 2 provides the overall pipeline of our method by an example of semantic relationship change. Additionally, we provide implementation details in Appendix A. We will also release our code to ensure reproducibility after the paper is accepted.

## 7 ETHICS STATEMENT

In this paper, we develop a model named SIMBIL for semantic image manipulation. From the experiments, we see that SIMBIL can effectively manipulate images according to user guidance. Our scene-graph driven editing technique not only make our system easy to use, but also help people with disabilities who cannot perform pixel-level image manipulation to manually segment, delete, and move the objects. However, just like other image editing approaches, SIMBIL can modify the content of the original images. Therefore, it is critical for practitioners to carefully review the modified images to avoid spreading misinformation. Additionally, we found SIMBIL-generated images are difficult to be recognized and located by exisiting image manipulation detection methods like IIDNet (Wu & Zhou, 2021). Thus, our internal learning method also provides a potential improvement direction for image manipulation detection, *i.e.*, how to develop a more robust framework that can recognize both the external learning manipulated images and the internal learning manipulated images.

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

# A    IMPLEMENTATION DETAILS

## A.1    SEGMENTATION MODULE

**CLEVR.** We apply the CLEVR framework (Johnson et al., 2017) to generate a dataset consisting of 1,000 images with object masks. Following (Dhamo et al., 2020), Mask R-CNN (He et al., 2017) is trained to classify the objects into 24 categories with 3 shapes (sphere, cylinder, cube) and 8 colors (blue, yellow, purple, green, red, gray, cyan, brown).

**Visual Genome.** We use MaskFormer (Cheng et al., 2021) pre-trained on COCO (Lin et al., 2014) for Visual Genome. In this paper, we focus on objects that the MaskFormer is trained to identify. However, these categories can be greatly expanded using methods like the transfer learning approach Mask$^X$ R-CNN (Hu et al., 2018) or segmenting entities based on natural language queries (Hu et al., 2016).

## A.2    RoI PREDICTION

**CLEVR.** As we mentioned in the main paper, the modified triplets, bounding boxes of reference objects and target objects are extracted from 6,688 images as our training data. 818 images are adopted as the validation set. We set the maximum number of the modified triplets of each image to 5. The mean absolute error (MAE) on validation set is 12.05±0.87 (computed over 5 runs), where the image size is $256 \times 256$. It should be mentioned that the gap between the predicted value and ground truth value does not necessarily indicate the predicted value is incorrect. Any points in the correct region can satisfy the relationships between objects. For instance, if a blue cube is on the right side of a red sphere, then any positions on the right side of the red sphere should be correct. Therefore, the accuracy of our model is underreported.

**Visual Genome.** Similar to CLEVR, we use 40,000 images for training and 2089 images for evaluation. The MAE on validation set is 77.84±1.25 (computed over 5 runs), where the image size is $512 \times 512$.

## A.3    BACKGROUND-GUIDED INTERNAL LEARNING

We use the encoder-decoder structure with skip connections (Ulyanov et al., 2018) for our inpainting model. The iteration number for each image is set to 2,000 and $\lambda$ in Eq. 5 is set to 0.1. For CLEVR, we constrain the average of each missing region by a single value. For Visual Genome, we constrain the average of missing regions row by row.

## A.4    MODIFICATIONS ON SCENE GRAPHS

To precisely reflect the modified relationships between objects, users may need to modify many graph edges. This process can be simplified by modifying only a few graph edges. For example, in our Visual Genome experiments, we observe that modifying a single triplet can also render plausible editing results, though it may bring randomness on RoI prediction.

Table 3: Ablation study of SIMBIL on CLEVR (256×256 image). DIP, guide, dilation denotes deep image prior, background-guided internal learning, and mask dilation respectively. See Section B for discussion.

| Method | All pixels | | | | RoI only | |
|---|---|---|---|---|---|---|
| | MAE ↓ | SSIM ↑ | LPIPS ↓ | FID ↓ | MAE ↓ | SSIM ↑ |
| DIP | 1.85 | 98.16 | 0.069 | 7.13 | 25.38 | 73.48 |
| DIP+guide | 1.73 | 98.19 | 0.069 | 8.07 | 19.85 | 75.97 |
| DIP+dilation | 1.72 | 98.26 | 0.065 | 7.21 | 22.02 | 75.60 |
| DIP+guide+dilation | **1.71** | **98.30** | **0.061** | **6.55** | **18.86** | **79.48** |

Table 4: Comparison of evaluating modifications A and B. See **Semantic correctness of edits** of Section B for details.

| | $l_1$ (RoI)↓ | SSIM(RoI) ↑ | | $l_1$(RoI) ↓ | SSIM(RoI)↑ |
|---|---|---|---|---|---|
| A | 18.86 | 79.48 | B | 41.04 | 65.56 |

## B  ADDITIONAL EXPERIMENTS

**Additional Qualitative Results.** Figures 7, 8, & 9 provide additional qualitative results that demonstrating SIMBIL outperforms prior work on Visual Genome to supplement the results from the main paper.

**Diverse Outputs.** As we discussed in Section A.2, a modified scene graph can point to different outputs with correct semantics based on the same input. We generate diverse outputs using the manipulated relationship's probability map shown in Figure 10.

**Semantic correctness of edits.** To illustrate that the incorrect semantics of manipulated images could be captured by our quantitative metrics in the main paper, we performed a comparison experiment in Table 4. We applied modified scene graph $A$ and modified scene graph $B$ on the same input and compared their outputs with the ground truth label of scene graph $A$ (note A≠B). We see the correctly modified image has significantly better results.

**Ablation Study.** Table 3 provides the ablation study of mask dilation, validating that the improvements of SIMBIL are based on both the background-guided mechanism and the mask dilation.

**Image Editing using Predicted Scene Graphs.** We performed an experiment using scene graphs predicted by F-Net (Li et al., 2018) in Table 5. Consistent with our experiments in Table 3, we performed four image manipulation operations on the test set, including object removal, object addition, object replacement, and relationship change. We see that using predicted scene graphs has a minor impact on the performance of image editing results compared to ground truth scene graphs. However, our method with predicted scene graphs still outperforms SIMSG (F-Net), which means ground truth scene graphs are not essential for the improvement of our model. For other baselines including GLIDE(Nichol et al., 2021) and BDM (Avrahami et al., 2022), since we use the same RoI as predicted by our model (otherwise, GLIDE and BDM require manually outlining RoI), the influence of predicted scene graphs should be the same.

**Iterations in optimization.** We compare the iterations of DIP and SIMBIL in Figure 11 to demonstrate the improvement of our background-guided mechanism. *E.g.*, in the bottom figure, the removed object is blue cylinder and DIP uses the color of purple cube which is far from the blue cylinder to fill in the missing part. In contrast, SIMBIL uses the color of grey background to fill in the missing part.

Table 5: Image editing using predicted scene graphs by F-Net (Li et al., 2018). GT denotes ground truth scene graphs. See **Image Editing using Predicted Scene Graphs** of Section B for details.

|  | $l_1 \downarrow$ | SSIM $\uparrow$ | $l_1$(RoI) $\downarrow$ | SSIM(RoI) $\uparrow$ |
|---|---|---|---|---|
| SIMSG (F-Net) | 8.52 | 96.92 | 27.81 | 74.74 |
| SIMSG (GT) | 8.30 | 96.97 | 24.59 | 78.61 |
| ours (F-Net) | 1.94 | 98.17 | 23.55 | 76.29 |
| ours (GT) | 1.71 | 98.30 | 18.86 | 79.48 |

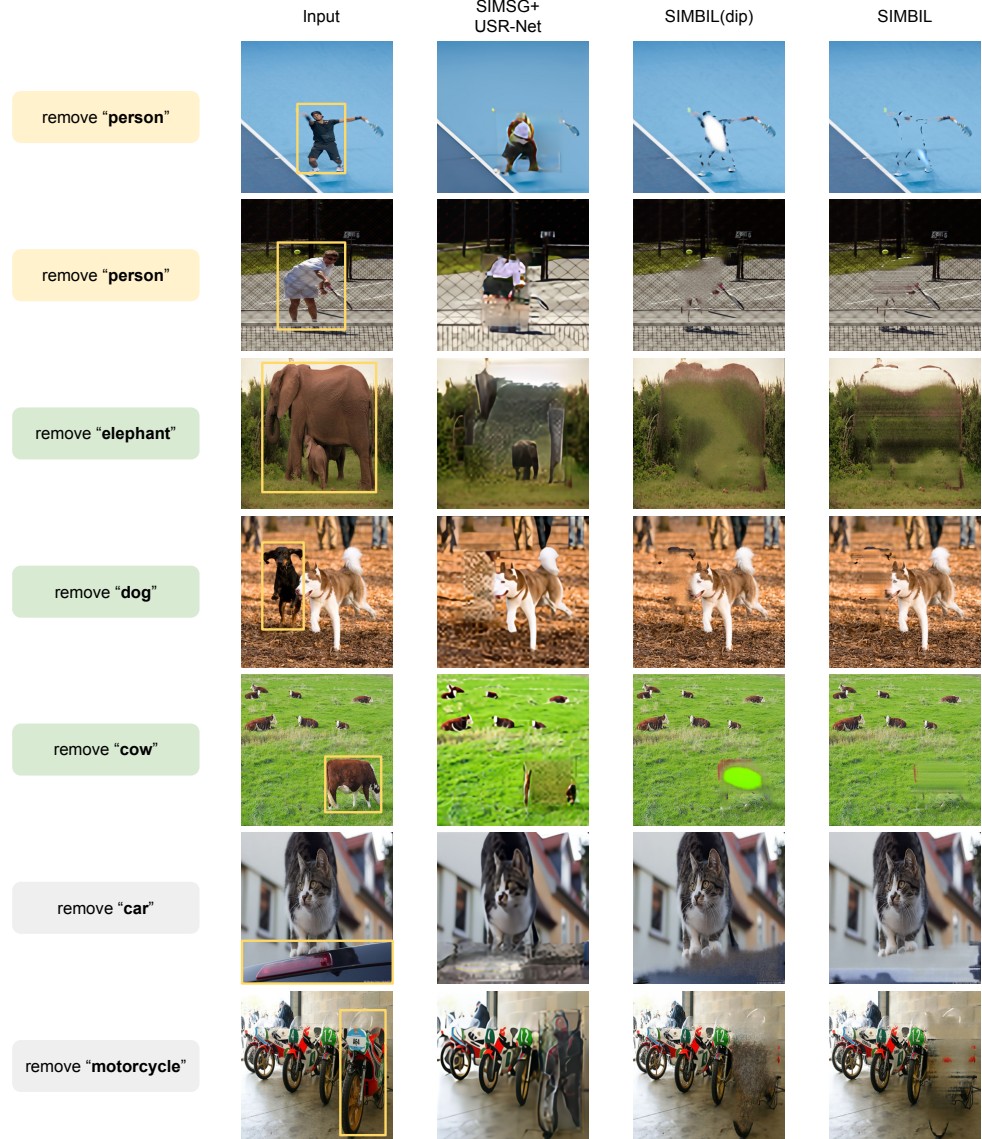

Figure 7: Object Removal. Target objects are outlined by light yellow bounding boxes. Consistent with our conclusion in Section4.3, our backgound-guided internal learning mechanism notably improves the inpainting results, especially when the removed object is large.

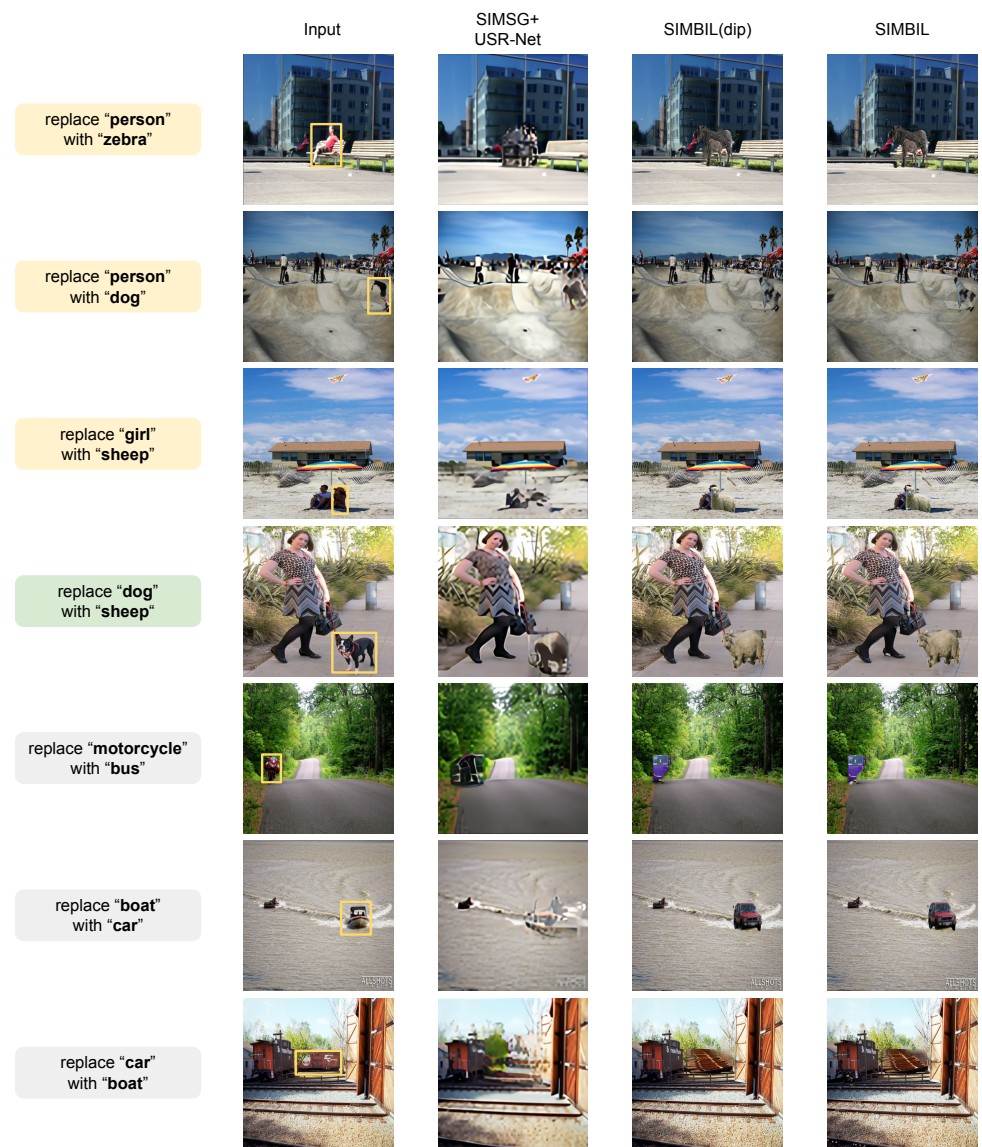

Figure 8: Object Replacement. Our method can effectively replace target objects according to editing commands.

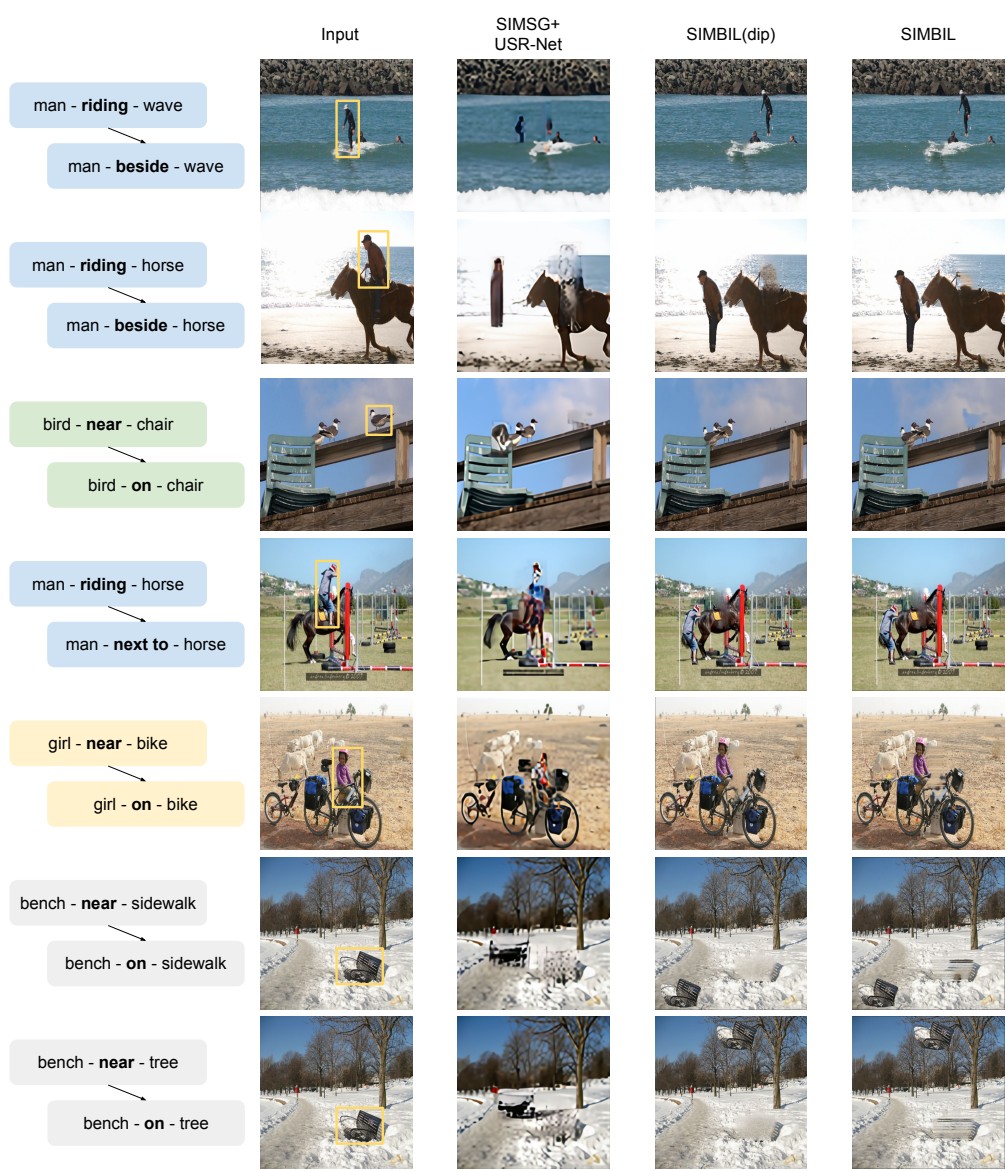

Figure 9: Relationship Change. SIMBIL puts target objects to reasonable positions according to different scene-graph modifications, supplementing discussion in Section 4.3.

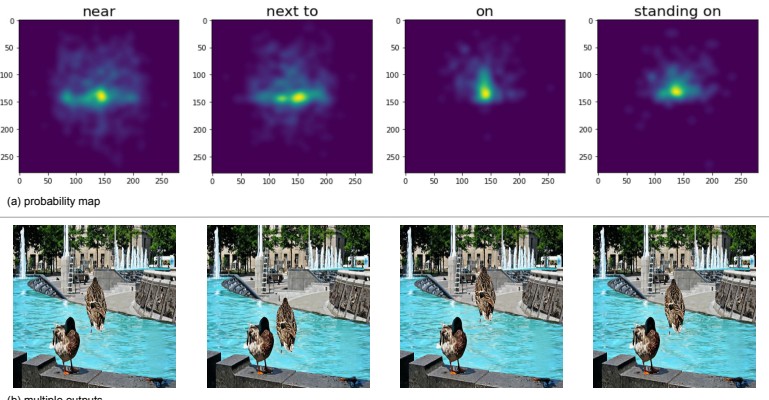

Figure 10: Top: probability map of selected predicates; Bottom: an example of diverse outputs, "bird-on-water".

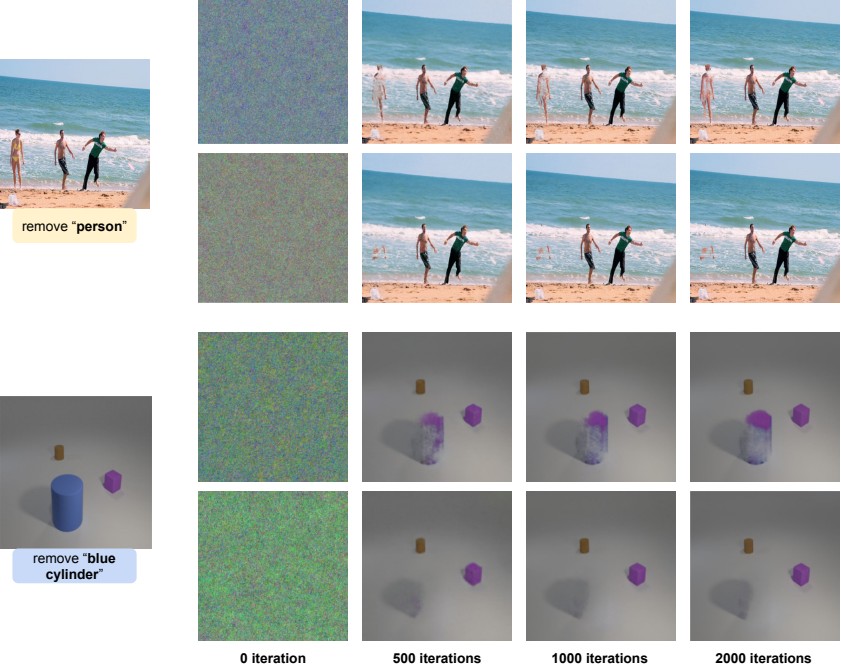

Figure 11: The optimization iterations of Deep Image Prior (Ulyanov et al., 2018) (top row of each image) and our method (bottom row of each image). We use background-guided mechanism to constrain the optimization of deep image prior and generate plausible results.

