# OpenReview forum: "Semantic Image Manipulation with Background-guided Internal Learning"
_ICLR.cc/2023/Conference — Submitted to ICLR 2023_

### Official Review · Reviewer_mRHW · 2022-10-19

**Confidence:** 4
**Correctness:** 4
**Technical Novelty And Significance:** 2
**Empirical Novelty And Significance:** 2
**Recommendation:** 5

**Clarity, Quality, Novelty And Reproducibility:**

This work gives us some new insights into image manipulation, which is interesting. The paper is well-organized and explained clearly.

**Strength And Weaknesses:**

### Strengths
1. Each of the four mentioned problems is well-defined and carefully processed in SIMBIL. Compared with existing methods, the method shows superior performance in semantic understanding and image manipulation.
2. The resource requirement of SIMBIL is more relaxed, without reliance on human annotations and external large datasets, thus being more user-friendly.
3. The paper is easy to follow and the system is easy to reproduce.

### Weaknesses
1. The comparison is not so convincing as the involved competing methods are not specifically trained on the datasets as SIMBIL does. It remains questionable whether SIMBIL will be still the best if other methods are well-finetuned or re-trained.
2. The visualized results of SIMBIL in Fig.4 and Fig.5 are far from satisfying. We observe obvious unappealing artifacts or the prediction is quite different from the ground truth, e.g., the replaced object in Fig.4 (the 2nd case) should be a ball instead of a cube.
3. The paper introduces a well-designed system, but the technical novelty is quite limited. The background-guided internal learning is quite ad-hoc for object removal. Besides, the system requires the GT scene graph for manipulation, which is impractical in real-world cases. Though scene graph generation methods could be adopted, we do not know the degree of performance drop so that the robustness of the system should be evaluated.
4. Some typos should be fixed. For example, the 2nd line 'Benefits from ... outline ...' should be 'Benefiting from ... outlining ...'.

**Summary Of The Paper:**

The paper proposes a semantic manipulation framework, combining low-level and high-level conditions, for object removal, object replacement, semantic relationship change and object addition. Quantitative and qualitative experiments validate the effectiveness of this design.

**Summary Of The Review:**

The paper develops a unified system for multiple image manipulation tasks, demonstrating its high flexibility. But the authors should compare different methods in a more rigorous manner, and the visual results should be further improved. More technical novelties are required.

---

> ### Author Response · Authors · 2022-11-19
> **Response to mRHW**
>
> We thank the reviewer for the comments, we appreciate their time and will use their suggestions to improve our paper.
>
> > The comparison is not so convincing as the involved competing methods are not specifically trained on the datasets as SIMBIL does. It remains questionable whether SIMBIL will be still the best if other methods are well-finetuned or re-trained.
>
> SIMBIL is a single image-based method. Therefore, it is not specifically trained on Visual Genome or CLEVR, as all the information it uses comes from the image at test time. Thus, although some methods in prior work like GLIDE and LDM are not trained on our datasets, we argue that our comparison is still fair since our approach does not require training on our datasets either.  Collecting data is expensive and should be considered when comparing methods. In other words, we are evaluating a setting where you are given some new data, and you can either use our approach or the pretrained models from prior work.  As our experiments show, our approach is far more effective in this setting, and would remain valuable even if an external learning method could boost performance if it was trained on the same domain.
>
> In addition, we do note that training on the same datasets does not mean a method would actually outperform our approach.  Specifically, we note that SIMSG was specifically trained on both CLEVR and Visual Genome. Despite this, our approach still outperforms SIMSG.
>
> > The visualized results of SIMBIL in Fig.4 and Fig.5 are far from satisfying. We observe obvious unappealing artifacts or the prediction is quite different from the ground truth, e.g., the replaced object in Fig.4 (the 2nd case) should be a ball instead of a cube.
>
> We note that our approach outperforms all other baselines in both automatic and human evaluations.  Thus, it is clear that prior work is even less satisfying, and that we do not get perfect results does not mean that our approach is not valuable and a contribution over prior work.  We also argue that if we were to get perfect performance to generate flawless images, there would be no need to study this problem further.
>
> In response to the exact example the reviewer mentioned, we believe the modified object in the 2nd case in Fig. 4 is a red sphere instead of a cube. It might be mistakenly recognized due to the small size of the red sphere.
>
> > The paper introduces a well-designed system, but the technical novelty is quite limited. The background-guided internal learning is quite ad-hoc for object removal.
>
> We note that the main contribution of our work is combining external and internal learning to balance the strengths and weakness of each, resulting in a system that significantly outperforms prior work.  Although the main motivation of our background-guided internal learning is designed to address challenges in object removal and relationship change, we demonstrate that it is effective on these tasks.  In particular, we note that we users judged our manipulated images to be more correct than prior work in up to 33% more cases, including far more costly external learning methods like GLIDE and Latent Diffusion.

---

> > ### Author Response · Authors · 2022-11-19
> > **Additional Response to mRHW**
> >
> > > The system requires the GT scene graph for manipulation, which is impractical in real-world cases. Though scene graph generation methods could be adopted, we do not know the degree of performance drop so that the robustness of the system should be evaluated.
> >
> > We thank the reviewer for these suggestions. We have updated our paper to include the experiments and discussion in Appendix B, Table 5. Specifically, we performed an experiment of image manipulation using scene graphs predicted by F-Net (Li et al., 2018). Consistent with our experiments in Appendix B, Table 3, we performed four image manipulation operations on the test set, including object removal, object addition, object replacement, and relationship change. We see that using predicted scene graphs has a minor impact on the performance of image editing results compared to ground truth scene graphs. However, our method with predicted scene graphs still outperforms SIMSG (F-Net), which means ground truth scene graphs are not essential for the improvement of our model. For other baselines including GLIDE(Nichol et al., 2021) and BDM (Avrahami et al., 2022), since we use the same RoI as predicted by our model (otherwise, GLIDE and BDM require manually outlining RoI), the influence of predicted scene graphs should be the same.
> >
> >
> >
> > **Image editing using predicted scene graphs by F-Net (Li et al., 2018). GT denotes ground truth scene graphs.**
> > |   Method   | MAE | SSIM | MAE (RoI) |  SSIM (RoI) |
> > |---------------|:----------:|-----------:| -----------:|-----------:|
> > | SIMSG (F-Net) | 8.52 | 96.92 | 27.81 | 74.74 |
> > | SIMSG (GT) | 8.30 | 96.97 | 24.59 | 78.61 |
> > | ours (F-Net)  |  1.94  |  98.17  |  23.55  |  76.29  |
> > | ours (GT)  | 1.71  |  98.30 | 18.86 | 79.48 |
> >
> >
> >
> > > Some typos should be fixed. For example, the 2nd line 'Benefits from ... outline ...' should be 'Benefiting from ... outlining ...'.
> >
> > We thank the reviewer for pointing this out and have updated our paper to fix this issue.

---

> > > ### Comment · Reviewer_mRHW · 2022-12-11
> > > **Response to Reviewers**
> > >
> > > Thanks for the careful response to my questions. The first and part of second and third concerns are well addressed. However, I still think the contributions (including the technical improvements and final performance) of this paper should be further enhanced to be accepted by ICLR.

---

> > > > ### Author Response · Authors · 2022-12-11
> > > > **Thank you Reviewer mRHW**
> > > >
> > > > Thank you for your time and effort, we appreciate it!  We would very much appreciate if the reviewer could provide some actionable feedback so that we can improve our paper, as we feel we have already provided both technical insights and significant performance gains over prior work, as we detail further below.
> > > >
> > > > The main contribution of our work is combining the benefits of internal and external learning methods, allowing our work to improve upon internal learning methods by allowing them to support high-level manipulation requests,  and improving on the generalization ability  of external learning methods (in terms of target data distribution as well as manipulation tasks as highlighted in Table 1) as our approach works well without target data for training.  This is a contribution not found in prior work, and one can consider this work even more important in light of recent advances in external learning methods.  In other words, it provides a method with strong generalization abilities, making it a good benchmark for these methods.
> > > >
> > > > Note that we did not simply adapt existing internal learning approaches, as we provided improvements upon those methods, such as our background-guided internal learning, that provided significant improvements upon prior work.  In particular, as shown in Table 2 of our paper, according to human judgements, our approach more accurately manipulates natural images (using Visual Genome) by 9-33%.  This represents is very strong performance gains above and beyond what is normally considered worthy of publication at venues like ICLR.  We would argue these gains alone are often used as justification for publication on their own, e.g., CLIP used large scale pretraining using methods that were very simple and already existed in prior work, but the performance gains coupled with thorough experiments were significant enough to justify publication.  However, we also include technical improvements over prior work, making our case for publication quite strong in this regard.
> > > >
> > > > Following the reviewer's suggestion, we also demonstrated that our approach can utilize predicted (rather than ground truth) scene graphs and still provide a significant gain over prior work.  In addition, we also highlighted in our responses to reviewer's  (e.g., Reviewer 8S2u) that our approach could be used to support text-based manipulations similar to those in GLIDE, as we could simply extract the relevant tuples to modify our scene graph using a dependency parser on the text query.
> > > >
> > > > In summary, our paper introduces new ideas that are relatively easy to implement while also providing a 9-33% gain in performance over prior work (even over methods that require orders of magnitude more computational resources to use).  As such, we would very much appreciate details either highlighting flaws in our experiments (as we are sure the reviewer agrees a 9-33% absolute performance gain is typically above the bar for publication) or citations to related work that would diminish the impact of our claimed technical contributions.  This would provide significant insight into the justification behind the reviewer's recommendation that is vital for improving our paper.  Thank you again!

---

### Official Review · Reviewer_Pno2 · 2022-10-24

**Confidence:** 4
**Correctness:** 2
**Technical Novelty And Significance:** 2
**Empirical Novelty And Significance:** 2
**Recommendation:** 5

**Clarity, Quality, Novelty And Reproducibility:**

Clarity: fair
Quality: fair
Novelty: poor
Reproducibility: fair

**Strength And Weaknesses:**

# Strength
1. This paper use scene graphs to build the relationship between objects. And the users can manipulate the image with the scene graphs, which is more flexible than the low-level control method.
2. The SIMBIL can automatically predict RoI for the object addition and semantic relationship change tasks. This strategy helps the users not select the specific regions of the object and improves the robustness of SIMBIL.
3. The background-guided internal learning is reasonable, which can help the complete content consistent with the background even with the large holes.

# Weakness
1. Is the correct scene graphs essential for the performance of image manipulation? The authors should compare the different scene graph prediction methods.
2. If the image has no multi-object, is the SIMBIL can work well? For example, there is a sky in an image, and the sky has many clouds, dose the scene graphs can build the relationship between these clouds?
3.  For object removal, the edge-connect is not the SOTA. The authors should compare the SOTA method [1,2].
4. The novelty of SIMBIL is lacking. The main difference between the proposed methods and SIMSG is background-guided internal learning. I think this is not enough for the ICLR. As I mentioned above, if SIMBIL uses the SOTA object removal method, I think the performance may be better.

[1] Suvorov, Roman, et al. "Resolution-robust large mask inpainting with fourier convolutions." Proceedings of the IEEE/CVF Winter Conference on Applications of Computer Vision. 2022.

[2] Li, Wenbo, et al. "MAT: Mask-Aware Transformer for Large Hole Image Inpainting." Proceedings of the IEEE/CVF Conference on Computer Vision and Pattern Recognition. 2022.



**Summary Of The Paper:**

This paper proposes SIMBIL to manipulate the image in high-level and low-level aspects. High-level manipulation requires less manual effort from the user compared to manipulating raw image pixels. The low-level internal learning approach is scalable to images of various sizes without reliance on external visual datasets for training. The experiments show the good performance of this paper.


**Summary Of The Review:**

See the weakness.

---

> ### Author Response · Authors · 2022-11-19
> **Response to Pno2**
>
> We thank the reviewer for the comments, we appreciate their time and will use their suggestions to improve our paper.
>
> > Is the correct scene graphs essential for the performance of image manipulation? The authors should compare the different scene graph prediction methods.
>
> We thank the reviewer for these suggestions. We have updated our paper to include the experiments and discussion in Appendix B, Table 5. Specifically, we performed an experiment of image manipulation using scene graphs predicted by F-Net (Li et al., 2018). Consistent with our experiments in Appendix B, Table 3, we performed four image manipulation operations on the test set, including object removal, object addition, object replacement, and relationship change. We see that using predicted scene graphs has a minor impact on the performance of image editing results compared to ground truth scene graphs. However, our method with predicted scene graphs still outperforms SIMSG (F-Net), which means ground truth scene graphs are not essential for the improvement of our model. For other baselines including GLIDE(Nichol et al., 2021) and BDM (Avrahami et al., 2022), since we use the same RoI as predicted by our model (otherwise, GLIDE and BDM require manually outlining RoI), the influence of predicted scene graphs should be the same.
>
> **Image editing using predicted scene graphs by F-Net (Li et al., 2018). GT denotes ground truth scene graphs.**
> |   Method   | MAE | SSIM | MAE (RoI) |  SSIM (RoI) |
> |---------------|:----------:|-----------:| -----------:|-----------:|
> | SIMSG (F-Net) | 8.52 | 96.92 | 27.81 | 74.74 |
> | SIMSG (GT) | 8.30 | 96.97 | 24.59 | 78.61 |
> | ours (F-Net)  |  1.94  |  98.17  |  23.55  |  76.29  |
> | ours (GT)  | 1.71  |  98.30 | 18.86 | 79.48 |
>
>
> > If the image has no multi-object, is the SIMBIL can work well? For example, there is a sky in an image, and the sky has many clouds, does the scene graphs can build the relationship between these clouds?
>
> From our understanding, the question the reviewer is asking is “does SIMBIL work well on images with a single object?” For this question, SIMBIL can work well because it does not need to consider complicated relationships between objects (e.g., images in Figure 5(c)).
>
> For images with multiple objects, scene graphs can build relationships between objects such as animals, cars, people, and so on (Figure 5(a), Figure5(b)). However, in the sky-clouds example, we note that annotations of Visual Genome always consider clouds in an image as a single object.  We ask the reviewer to please clarify their question if there has been any misunderstanding.
>
> > For object removal, the edge-connect is not the SOTA. The authors should compare the SOTA method [1,2].
>
> We have updated our paper to include the additional comparison to LAMA[1] and MAT[2] in Table 1. We observe that the methods suggested by the reviewer still cannot solve the domain-shift issue perfectly. However, we would like to point out that the task of our paper does not focus solely on image inpainting. Object removal is just a special case of scene-graph-based image editing, and we choose deep image prior as our backbone to solve the domain-shift issue. However, our approach would generalize if we were to replace deep image prior with other frameworks such as [1] or [2]. This does not affect the major contribution of our paper, which is combining high-level semantics with pixel-level manipulation (automatic Photoshop mechanism) for image editing.
>
> **Comparison to LAMA[1] and MAT[2] on CLEVR**
> |   Method          | MAE (RoI) |  SSIM (RoI) |
> |-------------------------------|:----------:|-----------:|
> | EdgeConnect | 29.26 | 66.70 |
> | LAMA  |  33.54  |  71.19  |
> | MAT  | 31.46  |  74.80   |
> | SIMBIL | 9.90  |  93.33  |
>
>
> [1] Suvorov, Roman, et al. "Resolution-robust large mask inpainting with fourier convolutions." Proceedings of the IEEE/CVF Winter Conference on Applications of Computer Vision. 2022.
>
> [2] Li, Wenbo, et al. "MAT: Mask-Aware Transformer for Large Hole Image Inpainting." Proceedings of the IEEE/CVF Conference on Computer Vision and Pattern Recognition. 2022.
>
> [3] Yikang Li, Wanli Ouyang, Bolei Zhou, Jianping Shi, Chao Zhang, and Xiaogang Wang. Factorizable net: an efficient subgraph-based framework for scene graph generation. In ECCV, pages 335–351, 2018.

---

> > ### Author Response · Authors · 2022-11-19
> > **Additional Response to Pno2**
> >
> > > The novelty of SIMBIL is lacking. The main difference between the proposed methods and SIMSG is background-guided internal learning. I think this is not enough for the ICLR. As I mentioned above, if SIMBIL uses the SOTA object removal method, I think the performance may be better.
> >
> > We note that the main difference between SIMSG is not just background-guided internal learning, as our entire framework is different from SIMSG. Specifically, SIMSG takes the original image and modified scene graph as input and trained their model by image reconstruction task. SIMSG can be considered as a GAN-based model. In contrast, our method takes modified scene graphs as input, and manipulates images at pixel level, which can be considered as an automatic Photoshop method. Thus, our method can preserve much better information of unmodified parts and is not limited to image size.
> >
> > As we discussed above, Object removal is just a special case of our scene-graph-based image editing task. Our major contribution is to combine high-level semantics with pixel-level image manipulation to solve the image editing task, and we apply an internal learning framework as our backbone to alleviate the domain shift issue. This is not tied to a single inpainting backbone, i.e., if a newly improved image inpainting method was developed to solve the domain-shift issue on open-ended images, our method can be further improved by using them without additional effort.  In addition, our results demonstrate that our approach is far more effective than prior work.  We argue this knowledge itself is quite valuable, as it would helps to understand the performance and limitations of research on this topic.

---

> > > ### Comment · Reviewer_Pno2 · 2022-12-09
> > > **Additional  Questions**
> > >
> > > As shown the Fig.5 (a), there is no zebra or dog in the original image, so dose the proposed  SIMBIL  blend the dog or zebra image into the target area with poisson blending? And other methods generate the content (zebra or dog) by themself?

---

> > > > ### Author Response · Authors · 2022-12-11
> > > > **Response to Reviewer Pno2**
> > > >
> > > > SIMBIL adopts a compositional image representation following Text2Scene (Tan et al., 2019), which means the dog and zebra used in SIMBIL are extracted from existing images. Baselines such as BDM (Avrahami et al., 2022) apply generative models to generate and blend the target object with the background.

---

### Official Review · Reviewer_8S2u · 2022-10-30

**Confidence:** 5
**Correctness:** 3
**Technical Novelty And Significance:** 1
**Empirical Novelty And Significance:** 1
**Recommendation:** 3

**Clarity, Quality, Novelty And Reproducibility:**

The authors are encouraged to improve the clarity and quality of this paper. The novelty of this paper is below the quality bar of an ICLR paper. As a straightforward improvement on DIP, this paper could be easily reproduced by modifying the DIP code.

**Strength And Weaknesses:**

**Strength**
- This paper makes attempts to edit an image with a modified scene graph.

- Experiments compare with many recent methods and a user study is given.

**Weakness**

- The contributions of this paper could be strengthened. For now, it is hard to find significant novelty and insights in the scene graph modification and RoI prediction part. As for the background-guided learning strategy, it simply separates the foreground and background and uses them for image generation. The overall technical novelty is weak.

- The deep image prior, as a single image-based method, performs much worse than the ones learned from a large-scale image dataset. Thus, it is not convincing that the authors mainly base on and compare with the deep image prior work in Sec. 3.3.

- The scene graph modification only contains four simple forms, making the use of a scene graph (as a scene graph can model very complicated scenes) questionable. Maybe a simple input triplet or a referring expression will work.

- I would like to see more relationship change results for complicated natural images.

- The visual results are unsatisfactory and contain noticeable artifacts. For example, the *remove* results are much worse compared with the reported results in state-of-the-art papers (the results of LDM are obviously wrong). The authors should check their experiment settings and make sure the compared results are correct.

- The organization and presentation of this paper could be improved. I am confused about the relationship between object removal and scene graph editing.


**Summary Of The Paper:**

This paper edits images with the guidance of a modified scene graph. Building on image deep prior, this paper takes the foreground and background separation into account for better synthesis quality. Experiments on CLEVR and visual genome illustrate the performance of the proposed method.

**Summary Of The Review:**

Considering the weak technical contributions and potential problematic implementation of other methods, I do not suggest this paper for acceptance.

---

> ### Author Response · Authors · 2022-11-19
> **Response to 8S2u**
>
> We thank the reviewer for their comments, we appreciate their time and will use their suggestions to improve our paper.
>
> > The contributions of this paper could be strengthened. For now, it is hard to find significant novelty and insights in the scene graph modification and RoI prediction part. As for the background-guided learning strategy, it simply separates the foreground and background and uses them for image generation. The overall technical novelty is weak.
>
> As we mentioned in Section 1, the major contribution of our work is to combine high-level semantics with pixel-level image manipulation. Specifically, we apply high-level semantics to RoI prediction, and then use an automatic PhotoShop mechanism to manipulate images at pixel-level. Compared to existing methods based on generative models, SIMBIL can generate high-resolution images while accurately preserving the original details of unmodified parts. In addition, we also purpose a background-guided internal learning mechanism to solve the domain-shift issue we encountered in our experiments. In contrast to deep image prior, our method utilizes the average value of the background pixels around the missing part as guidance as opposed to only relying on the implicit prior captured by the neural network parameterization, boosting performance. We note that these two contributions are unexplored by prior work, yet provide significant improvements as demonstrated by our experiments, and, thus, are considered both effective and novel. We would be happy to discuss any specific prior work suggested by the reviewer.
>
> > The deep image prior, as a single image-based method, performs much worse than the ones learned from a large-scale image dataset. Thus, it is not convincing that the authors mainly base on and compare with the deep image prior work in Sec. 3.3.
>
> According to our experiments, both methods have their own advantages and drawbacks. For example, internal learning generalizes well to open-ended images while external learning methods are likely to be overfitting on trained data. We discuss deep image prior in Section 3.3 because our internal learning method is based on deep image prior rather than external learning frameworks. Comparisons to external learning methods are presented in Section 4.2 and Section 4.3, and, thus, our experiments go beyond just comparing to deep image prior. We reiterate, that our experiment results show that our method also outperforms external learning baselines on both CLEVR (Table 1) and Visual Genome (Table 2), despite the fact our approach is not “trained” on this data. This is due, in part, to the fact that external learning methods suffer from domain shifts when applied to new settings, whereas our approach can be used off the shelf.  Collecting new data is expensive, and, thus, methods like ours are less expensive than external learning methods as they do not require having in-domain data to be effective.
>
>
> > The scene graph modification only contains four simple forms, making the use of a scene graph (as a scene graph can model very complicated scenes) questionable. Maybe a simple input triplet or a referring expression will work.
>
> As illustrated in Table 1, the fact that our approach can support four types of manipulation makes it more powerful with capabilities that are not supported in prior work.  In addition, we note that modifying a scene graph can be accomplished via an input triplet or referring expression, and need not actually modify the graph.  For example, an interface in Figure 1(c) could take the triplet (girl, next to, horse), and then check it against the nodes in a predicted scene graph, modifying those related to the girl and horse.   Even in the case where there were multiple horses in the image, we need only select the one that is closest to the girl by default, or the user could provide more triplets to ensure other relationships are retained or modified.  This highlights how the interface used by a human need not directly follow how the underlying model implements the modification.
>
> > I would like to see more relationship change results for complicated natural images.
>
> We have included additional relationship change results in Appendix B, Figure 9.
>
> > The visual results are unsatisfactory and contain noticeable artifacts. For example, the remove results are much worse compared with the reported results in state-of-the-art papers.
>
> We reiterate that we compared with many methods from the state-of-the-art and they performed worse than our approach according to both automatic metrics and a human study.  That said, our approach is not perfect, but we argue that research on this topic would not be necessary if methods were capable of making perfect images.  We also note that Visual Genome images are more complex with more objects on average than those in prior work, which are typically generated in restricted domains (e.g., just for faces, or simple scene images).

---

> > ### Author Response · Authors · 2022-11-19
> > **Additional Response to 8S2u**
> >
> > > The results of LDM are obviously wrong. The authors should check their experiment settings and make sure the compared results are correct.
> >
> > We have verified the output of LDM and find that our settings are both correct and effective when used in images from the same domain as LDM.  However, one of the key arguments in our work is that external learning methods like LDM suffer from domain shifts when supplied with images from outside their training images.  For example, LDM was trained on Places images, but Visual Genome contains more complex images with many objects. We note that in the publicly available github repository for MAT [1], they also found that the LDM inpainting model performs poorly in their experiments. While LDM may be able to improve performance by training on Visual Genome images, as noted earlier, our approach does not require collecting or training on Visual Genome, so our experiments are a valid illustration comparing our approach to the ability of LDM to generalize to new data.
> >
> > > The organization and presentation of this paper could be improved. I am confused about the relationship between object removal and scene graph editing.
> >
> > We thank the reviewer for the suggestions.  We note that we follow the image manipulation settings of prior work (e.g., SIMSG). For example, in object removal, the node and its corresponding edges of the target object are removed. We discuss how the manipulations are completed in Section 3.1 and Section 3.4. We are happy to include any additional suggestions.
> >
> > [1] Li, Wenbo, et al. "MAT: Mask-Aware Transformer for Large Hole Image Inpainting." Proceedings of the IEEE/CVF Conference on Computer Vision and Pattern Recognition. 2022.

---

### Official Review · Reviewer_BfVZ · 2022-11-04

**Confidence:** 4
**Correctness:** 3
**Technical Novelty And Significance:** 3
**Empirical Novelty And Significance:** 3
**Recommendation:** 6

**Clarity, Quality, Novelty And Reproducibility:**

The proposed method has a certain novelty and achieves promising results. However, insights and in-depth analytical discussions behind the proposed method are missing.
Besides, the writing of the article needs to be improved. There are some misinterpretations of basic knowledge, some sentences that are difficult to understand, and some self-contradictory descriptions.


**Strength And Weaknesses:**

Strength
1. The idea of manipulating image by modifying its scene graph is new and has potential.
2. The proposed background-guided internal learning makes sense and alleviates the drawbacks of existing schemes.
3. The results on the validation benchmark are promising and surpass prior semantic image manipulation methods.


Weaknesses
1. There are some inaccuracies or self-contradictions in the description of high-level ideas of the proposed method. For example,
(a) In Sec.1, the authors state “low-level manipulation spans image inpainting, object removal… do not need to understand the semantic meaning of an image”, which should be wrong. First, addressing tasks like image inpainting and object removal via neural networks requires semantic information. For example, for inpainting, it is necessary to understand the semantic information around the missing area to perform meaningful completion.
Similarly, before removing an object, it is necessary to know its semantic information to accurately segment the entire object. Second, the word “low-level” may be confusing here. “low-level” usually refers to tasks that are not related to image content/semantics, such as image denoising and deblurring. These tasks do not need to know what the image content is. Therefore, it is more accurate to use words like "pixel-level manipulation" rather than “low-level manipulation”.

(b) In Sec.2, the authors state “image manipulation can be seen a special case of image synthesis”, which may be inaccurate. Image manipulation not only includes synthesis, many other operations, such as image enhancement, cropping, and denoising, are all image manipulations. Besides, some statements like “Unlike methods based on generative networks, SIMBIL … directly manipulating raw pixels” are difficult to understand (as SIMBIL also uses background-guided internal learning to generate pixels).

(c) In Sec.3, the authors state “we perform four tasks of semantic manipulation: object addition, …, and object removal”, which conflicts with the statement in (a) above because “object removal” becomes a “semantic manipulation” here.

2. It is not clear how the proposed components form object replacement flow. Specifically, whether the proposed framework adding the target object before or after completing the missing background?

3. The rationale for using RoI Prediction for Relationship Change may be further discussed. Specifically, for object Relationship Change, modifying the scene graph requires a huge effort because many graph edges need to be modified (e.g., in Fig.2, moving the blue cylinder needs to change five edges carefully). Instead, the operation of directly dragging/resizing the segmented object to the specified position should be faster and more accurate (e.g., in Fig.2, users may directly drag the blue cylinder to a certain position in a precise manner). After that, the scene graph may be updated according to the user interactions in some way.


4. Since the discussed limitations like artificial boundary and biased background are also appear in other tasks like image inpainting / generation. More task-specific limitations may be discussed. For example, as the proposed methods require scene graph as an input. In practice, a complete scene graph requires huge efforts to create, especially scenes with lots of objects. Hence, how robust the proposed method when inputting an incomplete scene graph?


**Summary Of The Paper:**

The paper proposes a new framework, named SIMBIL, for semantic image manipulation. The framework contains three components, including a segmentation module to extract the target object according to the modified scene graph, a RoI prediction module to determine the new location of the target object, and a background-guided internal learning to complete the “hole” in the original location of the target object.

**Summary Of The Review:**

The paper proposed an interested method with great results. But the writing of the paper should be improved, and more analysis may be included. I would like to see in the rebuttal (a) the authors' answers to the above questions and (b) the authors' ideas for improving the manuscript to remove incorrect or contradictory descriptions, which determine the final score.

---

> ### Author Response · Authors · 2022-11-19
> **Response to BfVZ**
>
> We thank the reviewer for the comments, we appreciate their time and will use their suggestions to improve our paper.
>
>
> > There are some inaccuracies or self-contradictions in the description of high-level ideas of the proposed method. For example, (a) In Sec.1, the authors state “low-level manipulation spans image inpainting, object removal… do not need to understand the semantic meaning of an image”, which should be wrong. First, addressing tasks like image inpainting and object removal via neural networks requires semantic information. For example, for inpainting, it is necessary to understand the semantic information around the missing area to perform meaningful completion. Similarly, before removing an object, it is necessary to know its semantic information to accurately segment the entire object. Second, the word “low-level” may be confusing here. “low-level” usually refers to tasks that are not related to image content/semantics, such as image denoising and deblurring. These tasks do not need to know what the image content is. Therefore, it is more accurate to use words like "pixel-level manipulation" rather than “low-level manipulation”.
>
> We thank the reviewer for pointing out this statement. We agree that segmenting target objects according to text or scene graph does need the model to understand high-level semantics of the image. Some prior work (e.g., [1, 2]) consider image inpainting as a low-level vision task because they take object masks as input (In contrast, SIMBIL takes modified scene graphs as input). To avoid confusing statements, we have updated our paper to replace low-level manipulation with pixel-level manipulation.
>
> > In Sec.2, the authors state “image manipulation can be seen a special case of image synthesis”, which may be inaccurate. Image manipulation not only includes synthesis, many other operations, such as image enhancement, cropping, and denoising, are all image manipulations.
>
> While some manipulations do require synthesizing new objects, and thus could be considered (partly) an image generation problem, we do agree that it may also refer to other operations such as cropping, flipping, etc. We have updated our paper to better represent this broader view of the topic.
>
> >Besides, some statements like “Unlike methods based on generative networks, SIMBIL … directly manipulating raw pixels” are difficult to understand (as SIMBIL also uses background-guided internal learning to generate pixels).
>
> Although SIMBIL also uses background-guided internal learning to generate pixels, this does not require training data like general adversarial networks. We follow the arguments in Text2Scene (Tan et al., 2019), which considers the inpainting module as post-processing and claims that their method does not use generative adversarial networks.  However, we are happy to include the reviewer’s feedback to improve our paper and help clarify these points.
>
> > In Sec.3, the authors state “we perform four tasks of semantic manipulation: object addition, …, and object removal”, which conflicts with the statement in (a) above because “object removal” becomes a “semantic manipulation” here.
> It is not clear how the proposed components form object replacement flow. Specifically, whether the proposed framework adding the target object before or after completing the missing background?
>
> Existing methods [1, 2] consider image inpainting as a low-level vision task since the inputs to these models are object masks. In contrast, the inputs to our model are modified scene graphs and the model needs to figure out object masks by itself. This is the reason that we claim object removal as a semantic manipulation task in our paper. We discussed the manipulation for object addition in Section 3.4. Specifically, we adopt compositional scene representations for object addition and replacement following Text2Scene (Tan et al., 2019). Therefore, SIMBIL adds the target object before completing the missing background.
>
>
> [1] Zhao, Guoping, et al. "A deep cascade of neural networks for image inpainting, deblurring and denoising." Multimedia Tools and Applications 77.22 (2018): 29589-29604.
> [2] Ulyanov, Dmitry, Andrea Vedaldi, and Victor Lempitsky. "Deep image prior." Proceedings of the IEEE conference on computer vision and pattern recognition. 2018.

---

> > ### Author Response · Authors · 2022-11-19
> > **Additional Response to BfVZ**
> >
> > > The rationale for using RoI Prediction for Relationship Change may be further discussed. Specifically, for object Relationship Change, modifying the scene graph requires a huge effort because many graph edges need to be modified (e.g., in Fig.2, moving the blue cylinder needs to change five edges carefully). Instead, the operation of directly dragging/resizing the segmented object to the specified position should be faster and more accurate (e.g., in Fig.2, users may directly drag the blue cylinder to a certain position in a precise manner). After that, the scene graph may be updated according to the user interactions in some way.
> >
> > We thank the reviewer for this suggestion. We have updated our paper to include the discussion in Appendix  A.4. Specifically, users do need to modify many graph edges if they would like to get a very precise position for the target object. However, this process can be simplified by modifying only a few graph edges. For example, in our Visual Genome experiments in Section 4.3, we observe that modifying a single triplet can also render plausible editing results, though it may bring additional randomness to RoI prediction.
> >
> > > Since the discussed limitations like artificial boundary and biased background are also appear in other tasks like image inpainting / generation. More task-specific limitations may be discussed. For example, as the proposed methods require scene graph as an input. In practice, a complete scene graph requires huge efforts to create, especially scenes with lots of objects. Hence, how robust the proposed method when inputting an incomplete scene graph?
> >
> > We thank the reviewer for these suggestions. We have updated our paper to include the experiments and discussion in Appendix B, Table 5. Specifically, we performed an experiment of image manipulation using scene graphs predicted by F-Net (Li et al., 2018). Consistent with our experiments in Appendix B, Table 3, we performed four image manipulation operations on the test set, including object removal, object addition, object replacement, and relationship change. We see that using predicted scene graphs has a minor impact on the performance of image editing results compared to ground truth scene graphs. However, our method with predicted scene graphs still outperforms SIMSG (F-Net), which means ground truth scene graphs are not essential for the improvement of our model. For other baselines including GLIDE(Nichol et al., 2021) and BDM (Avrahami et al., 2022), since we use the same RoI as predicted by our model (otherwise, GLIDE and BDM require manually outlining RoI), the influence of predicted scene graphs should be the same.
> >
> >
> > **Image editing using predicted scene graphs by F-Net (Li et al., 2018). GT denotes ground truth scene graphs.**
> > |   Method   | MAE | SSIM | MAE (RoI) |  SSIM (RoI) |
> > |---------------|:----------:|-----------:| -----------:|-----------:|
> > | SIMSG (F-Net) | 8.52 | 96.92 | 27.81 | 74.74 |
> > | SIMSG (GT) | 8.30 | 96.97 | 24.59 | 78.61 |
> > | ours (F-Net)  |  1.94  |  98.17  |  23.55  |  76.29  |
> > | ours (GT)  | 1.71  |  98.30 | 18.86 | 79.48 |

---

### Author Response · Authors · 2022-12-08
**Further Discussion**

We thank the reviewers for their comments and suggestions. We would like to know if there are other questions or inquiries about our paper, and we are more than happy to address any additional comments. We appreciate your time and look forward to your update.

---

### Decision · Program_Chairs · 2023-01-20

**Decision:**

Reject

**Justification For Why Not Higher Score:**

The technical issue with limited novelty makes the main concern of the current work.

**Justification For Why Not Lower Score:**

N/A

**Metareview: Summary, Strengths And Weaknesses:**

This submission receives 1 slightly positive review and 3 negative reviews. The raised issues include unclear technical presentation, limited contributions to existing technical modules, unmotivated scene graph constructions, and unsatisfied experimental results. Although the authors try to address these issues during the rebuttal phase, the novelty concern still exists, and the AC feels the current work is not ready for reporting in this venue. The authors shall take the suggestions to further improve the current submission and welcome to the next venue.